# Status Epilepticus in Post-Transplantation Hyperammonemia Involves Careful Metabolic Management

**DOI:** 10.3390/life12101471

**Published:** 2022-09-21

**Authors:** Vikram Venkata Puram, Brent Berry, Malik Ghannam, Yuka Furuya

**Affiliations:** 1Department of Internal Medicine, Stanford University, Stanford, CA 94304, USA; 2Department of Neurology, University of Minnesota, Minneapolis, MN 55414, USA; 3Department of Neurology, University of Iowa Hospitals and Clinics, Iowa City, IA 52242, USA; 4Department of Pulmonary Medicine, Allergy Critical Care and Sleep Medicine, University of Minnesota, Minneapolis, MN 55414, USA

**Keywords:** post-transplantation hyperammonemia, status epilepticus, encephalopathy, lung transplantation

## Abstract

Hyperammonemia is a condition that may result after solid organ transplantation, particularly lung transplantation. However, it is very uncommon for this presentation to occur more than 30 days post-transplantation. Hyperammonemia and the resulting encephalopathy typically manifest with altered sensorium, a clinical situation which is not often included in the differential diagnosis of presumed nonconvulsive status epilepticus (NCSE). Seizures are common among this subset of patients with hyperammonemia and may be refractory to traditional treatments. Evidence of elevated intracranial pressure by invasive monitoring and neuroimaging findings of diffuse cerebral edema are commonly reported. Here we examine the therapeutic importance of identifying the specific cause of hyperammonemic encephalopathy, a condition which may result in status epilepticus and ultimately cerebral edema or even brain death.

## 1. Introduction

Altered sensorium, among other neurologic sequalae, may occur as a result of solid organ transplantation. In general, neurologic causes of altered sensorium are often stratified into distinct categories such as toxic insults, metabolic disturbances, central nervous system lesions, or infectious etiology [1]. In the clinical management of these post-transplant patients with an unclear cause for altered sensorium, a thorough exclusion of central nervous system infection (lumbar puncture), structural abnormalities (brain imaging), seizure/electrical activity (electroencephalogram), and various laboratory studies (particularly serum ammonia levels for hepatic encephalopathy evaluation) is routinely performed.

Elevated serum ammonia is a well-known cause of status epilepticus because ammonia easily crosses the blood–brain barrier and may conjugate with α-ketoglutarate to form glutamate. This leads to brain damage and the precipitation of seizures given the excitatory activity of glutamate in the synaptic membrane [2]. The phenomenon of hyperammonemia in immunosuppressed post-transplant patients is not well-understood; however, more recently, several studies have detailed a relationship between hyperammonemia and *Ureaplasma* infection in immunosuppressed patients, including patients with lung transplantation [3,4]. Indeed, *Ureaplasma* species infection in immunosuppressed patients has been shown to cause rising serum ammonia levels causing dramatic CNS damage resulting in seizures and cerebral edema [4].

This report outlines a novel case of hyperammonemia occurring more than 30 days after lung transplantation in a patient who was found to have refractory convulsive and non-convulsive status epilepticus in the absence of *Ureaplasma* spp. infection. Although immediate administration of ammonia-lowering treatments and appropriate antibiotic therapy resulted in a temporary improvement of hyperammonemia, the patient unfortunately died.

## 2. Scenario

The patient was a 44-year-old male with a past medical history of idiopathic pulmonary fibrosis (IPF) well-controlled on at-home oxygen therapy, gastrointestinal reflux disease, and depression. He presented to a local hospital with worsening shortness of breath of one month, acutely worsened over the prior week. The patient underwent high-resolution computed tomography (CT) of the chest which showed pulmonary fibrosis similar in severity to prior CT scans but with new left lung-predominant ground glass opacities, consistent with acute exacerbation of chronic interstitial lung disease (Figure 1).

The differential diagnosis at the time included atypical pneumonia and pulmonary edema; however, bronchoalveolar lavage revealed negative bacterial cultures and fungal stains. There was no evidence of malignancy on imaging. The patient was treated with methylprednisolone, nebulized ipratropium bromide/albuterol, Ellipta (fluticasone furoate 100 mcg and vilanterol), ceftriaxone, and azithromycin. He was transferred to the ICU for worsening respiratory status and started on BiPAP, after which his respiratory status stabilized. He was transferred to our hospital for emergent evaluation for potential lung transplantation. His diagnosis of IPF was one year prior to this admission and this was his first acute exacerbation. He had been seen by his outpatient pulmonologist one month prior to this admission and was started on prednisone 20 mg daily for worsening dyspnea.

After transfer to our hospital, the patient required escalation to the ICU on hospital day 3 (HD3) and required intubation for worsening hypoxia not supported by BiPAP. His respiratory status continued to worsen and Veletri (epoprostenol) therapy was initiated. He was paralyzed on HD4, and then proned for 12 h (which paradoxically worsened the patient’s respiratory state). The patient’s pO2 had remained stable with a P/F ratio of 65; however, his pCO2 was persistently elevated in the 80s–90s. Multiple ventilator modes had been trialed without improvement. At this point, vasopressors were initiated, and combination therapy with a furosemide drip and metolazone was initiated for worsening pulmonary edema. Due to lack of improvement and further maneuvers available to improve oxygenation/ventilation, the patient was cannulated for veno-venous extracorporeal membrane oxygenation (VV ECMO). He was cannulated on HD5 as a bridge to lung transplant. Although his hospital course was complicated by multiple PEA arrests, he eventually received bilateral lung transplantation and subsequent ECMO decannulation on HD40.

After a successful transplant procedure, the patient was functionally improving, tolerating G-tube feeds, and had a decreasing oxygen requirement. He began to resume having meaningful interactions with his family and care team. His hospital course at this point was further complicated by multiple infections (including polymicrobial bacteremia and fungemia), intermittent confusion, delirium, anasarca, and persistent bilateral pneumothoraces of unclear etiology. Thirty days after transplantation, on HD70, the patient’s kidney function significantly worsened (baseline creatinine of 0.6–0.8 mg/dL with GFR > 90) with creatinine elevated to 1.44 mg/dL, GFR reduced to 58, and BUN elevated markedly) (Figure 2).

This acute kidney injury (AKI) was likely due to decreased renal perfusion from mild hypotension (blood pressure in the 90 s/60 s mmHg) and possibly exacerbated by tacrolimus toxicity (tacrolimus levels found to be elevated between 19 and 25). The patient also had an elevated level of voriconazole to 8.6 on HD 67, which was three days before AKI onset. Due to drug–drug interactions, levels of both drugs were monitored closely, and tacrolimus was down-titrated. With only a transient increase in trough levels, it was considered unlikely that tacrolimus was contributing to the patient’s acute change in mental status as the symptoms of tacrolimus-associated neurotoxicity are known to be reversible in most patients upon dose-reduction or drug discontinuation [5]. With respect to new-onset AKI, no contrast dye or excessive NSAIDs were used. On HD71 and HD72, the patient was improving slowly and was weaned off the ventilator; however, on HD73 he had to go to the operating room for sternal wound debridement and vacuum-assisted wound closure after which he was readmitted to the ICU.

The patient had elevated liver transaminases throughout his admission, with ALTs rising above 1000 IU/L and ASTs rising above 500 IU/L on numerous occasions, particularly around HD10. His alkaline phosphatase remained in the 200–300 IU/L range throughout his admission and his total bilirubin was elevated to above 10 mg/dL. He was thought to have cholestasis of critical illness but an ultimate diagnosis for these abnormal laboratory values was not clear. Prior to HD40, INR had remained below 1.5; however between HD40 and HD60, INR was elevated to an average of ~3.0 before decreasing slightly to INR of ~2.0 from HD60 and onwards. Bilirubin remained between 1 and 2 mg/dL until HD70 after which it began to rise precipitously to a peak of 18 mg/dL on HD 85. Ammonia was checked several times and was low prior to HD70 (cooled ammonia was probed to minimize possibility of false positive reading).

On HD80, after several days of increasing somnolence and ammonia levels rising above 250 µ/dL, the patient was started on hemodialysis and soon after his first run of dialysis, he was found to have a positive oculocephalic reflex “doll’s eyes test” and thus a stroke code was called. He was examined by neurology and was found to have minimal responses to sternal rub. His eyes were conjugate with pupils 8 mm and equal and reactive to light. Oculocephalics, corneal, and gag reflexes were intact at the time. Roving eye movements were not observed though the patient’s eyes had deviated to the right. The patient had no abnormal movements, no response to painful stimuli, and was found to be diffusely hyporeflexic yet symmetric in biceps, brachioradialis, patellar, and achilles. Head CT without contrast was obtained and found to be unremarkable. Consideration was given to stroke, but this was felt far less likely given the normal head CT. Concern for seizure was also raised although no acute treatment was given at the time; however, EEG was later ordered and revealed frequent bilateral occipital epileptiform activity (Figure 3).

During EEG monitoring, the patient had a total of 19 recorded seizures. Three of the seizures were prolonged (lasting 15–30 min each) and had clinical signs of head jerking and facial twitching. All seizures showed bilateral occipital onset with rhythmic and repetitive sharps which evolved in both frequency and amplitude with right occipital predominance. There were 16 subclinical seizures recorded with bilateral occipital activities, and also with right occipital predominance. These findings were consistent with status epilepticus and the patient was loaded with IV levetiracetam and then started on a midazolam drip with burst suppression. Overnight the patient proceeded into non-convulsive status epilepticus (NCSE) and required IV lacosamide. He was on 10 mg/hr midazolam which was then weaned abruptly the following day. That night the patient emerged into convulsive status epilepticus and was re-bolused with midazolam, which was then increased serially up to 50 mg/hr with no effect over the next two hours. He was then loaded with fosphenytoin which for several hours resulted in dissipation of convulsive activity and <2.5 Hz Bi-PLEDs (bilateral-type periodic epileptiform discharges). Two hours later, the patient returned to NCSE with 1 Hz Bi-PLED activity noted. Ketamine was loaded at this point and all seizure activity stopped. EEG was noted flatlining and a head CT was obtained revealing profound edema (Figure 4). The patient was examined and pronounced brain-dead.

## 3. Discussion

Behavioral disturbances and changes in consciousness are common occurrences in patients admitted to the intensive care unit (ICU), and can be caused by several factors such as medical conditions, medication side effects, and substance use [1]. Patients with these symptoms required immediate neurological evaluation and subsequent management; however, in severe situations, symptom management must take priority over determining the exact cause. The term status epilepticus (SE) describes the brain’s state when in persistent seizure. SE is a life-threatening condition with a very high mortality rate (estimated at 7.6% to 43%), especially when early identification and treatment is not performed in a timely manner [2]. SE has a wide variety of known causes; however, metabolic abnormality, especially hyperammonemia, is a rare etiology. Hyperammonemia may present with a broad range of symptoms including hypotonia, seizures, emesis, and changes in neurologic function such as stupor, disorientation, and tremors, and may even result in coma. Serum ammonia levels greater than 200 µmol/L have been associated with both brain edema and herniation [3,4,6,7,8,9,10], This article seeks to highlight a unique case of hyperammonemia in a patient post-lung transplantation, manifesting as abrupt neurologic disturbance and subsequent SE.

Both serum ammonia and inflammation play an important role in hepatic encephalopathy. Although hyperammonemia due to severe hepatic dysfunction is a well-documented phenomenon, hyperammonemia in the context of only marginally abnormal liver function tests (LFTs) represents a particular diagnostic difficulty [11,12,13,14,15]. When there is suspicion that hyperammonemia has an extra-hepatic cause, diagnostic workup and treatment for a metabolic disorder (such as excessive amino acid load/increased catabolism) alongside concurrent evaluations for neoplasms, circulatory shock with or without portosystemic shunt, UTI with urease-producing bacteria, as well as side effects of chemotherapeutic or anticonvulsant drugs are often implemented [16,17,18,19,20,21]. Laboratory tests for enzyme deficiencies were considered low on the differential and were not conducted in the case. Of note, *Ureaplasma* spp., which have been linked with hyperammonemia, particularly in lung transplant patients, were tested for and not found to be present.

Regardless of the exact cause of hyperammonemia, a rapid reduction in serum ammonia levels is of utmost importance. Next steps include stringent protein restriction in the diet as well as introduction of glucose- or dextrose-based solutions in order to encourage anabolism. Supportive treatments also include close management of hydration, nutritional, mineral (calcium), and electrolyte status. Nitrogen scavenging drugs, including sodium benzoate and sodium phenylacetate, may also be utilized, both of which are excreted in the urine.

When the rate of ammonia removal is less than ammonia production, the role of dialysis becomes significant, as seen in this patient’s case. Unfortunately, dialysis may sometimes cause rebound increases in ammonia through increased catabolism from serum nutrient removal. Given the high levels of serum ammonia and due to the onset of AKI in our patient, hemodialysis was initiated. Some advocate the use of continuous renal replacement therapy (CRRT) which may have been more beneficial in our patient; however, the consideration of patient hemodynamics becomes more important in these situations.

Additionally, Shah et al. have reported that ketamine may be useful in treating hyperammonemic seizures and other researchers have provided evidence that its use may be warranted in encephalopathy [22,23,24,25]. If hyperammonemia is recognized as a cause of seizures, using ketamine early in the seizure course—perhaps even before other anti-epileptic drugs (AEDs) are initiated—may be justifiable, though this is not currently included in any mainline SE treatment algorithm, a clear barrier to the early use of ketamine, which also occurred in this patient case.

The unfortunate circumstances of this patient’s case provide several learning points: (1) In an otherwise unexplained encephalopathy mimicking NCSE, plasma ammonia should be included in the initial laboratory panel due to its diagnostic and therapeutic implications. Often, laboratory test results for genetic etiology are not quickly available in the setting of acute metabolic crisis; (2) The standard therapeutic guidelines, other than in cases of adult-onset type II citrullinemia which suggest dextrose infusion in primary hyperammonemic encephalopathy, should be reviewed and individualized on a case-by-case basis; in particular, carbohydrate restriction should be considered in the rare cases of adult-onset type II citrullinemia; (3) Apart from the classical phenotype of spastic paraparesis, an additional feature of hyperargininemia includes episodic hyperammonemic encephalopathy [23]. Overall, we emphasize the importance of recognizing hyperammonemic encephalopathy as an imitator of NCSE, as hyperammonemia is an eminently treatable medical emergency and would prove fatal if not appropriately treated. A timely review of the amino acid profile can guide therapeutic decisions at the time of metabolic crisis.

## 4. Conclusions

Severe hyperammonemia is associated with high morbidity and mortality. When hyperammonemia is of extra-hepatic etiology, prompt and early diagnosis followed by appropriate intervention is paramount but challenging. This patient case highlights several possible causes of extra-hepatic hyperammonemia that are unlikely when considered separately, but highly possible when considered together. In hindsight, serial levels of ammonia including a baseline level would have been very helpful in managing the patient; whether it would have altered the outcome, though, remains uncertain. From the lessons of our case, we recommend that serum levels of ammonia should be included in the initial diagnostic workup of patients with risk factors for hyperammonemia, especially if the patient’s presentation involves altered consciousness and convulsion/seizure.

## Figures and Tables

**Figure 1 life-12-01471-f001:**
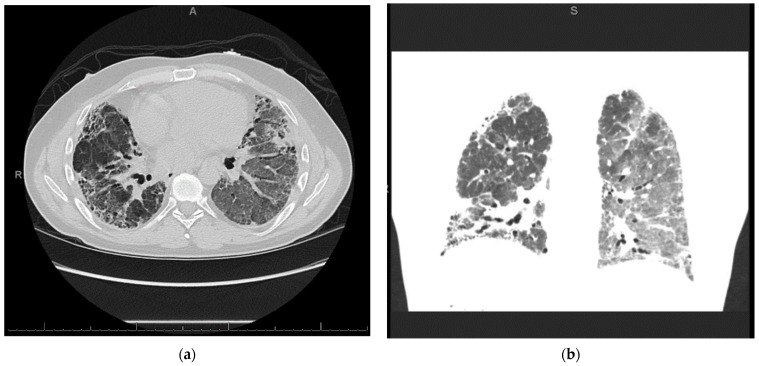
High-resolution CT chest: (**a**) Interlobular septal thickening with traction bronchiectasis and subpleural honeycombing, most prominent in the lower lobes. (**b**) Biapical subpleural fibrosis and pleural calcifications are seen at the left lung apex. New ground-glass opacities throughout the left lung with mild ground-glass opacities in the right lower and middle lobes. No endobronchial lesion. No pleural effusion.

**Figure 2 life-12-01471-f002:**
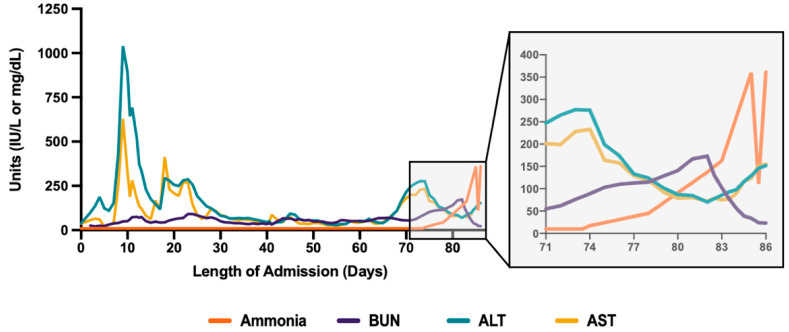
Ammonia, blood urea nitrogen, ALT, and AST levels across the patient’s admission. The patient underwent lung transplantation on HD40. Dialysis was initiated on day of stroke code on HD80. Patient passed away on HD86.

**Figure 3 life-12-01471-f003:**
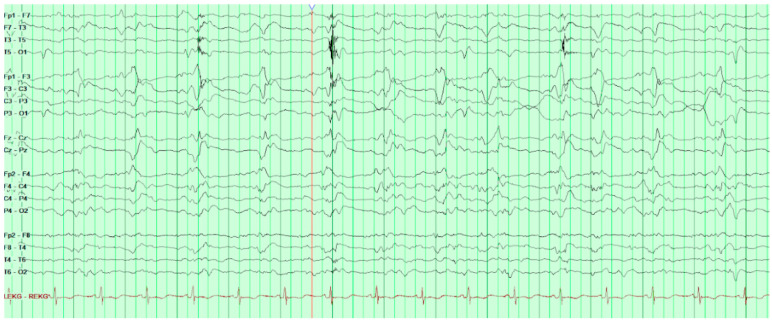
Background activities of this EEG consisted of moderate to severe generalized slowing with mixed theta and delta activities throughout the recording. The patient remained unresponsive throughout the recording. There were frequent bilateral occipital epileptiform activities noted. These findings are consistent with status epilepticus.

**Figure 4 life-12-01471-f004:**
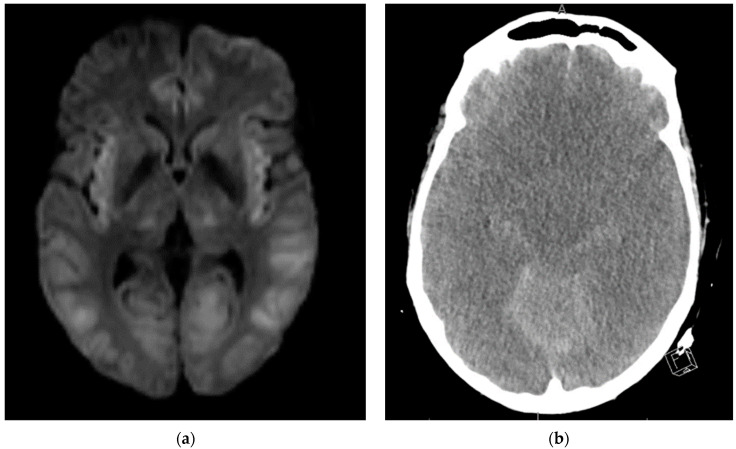
(**a**) Brain MRI DWI axial sequence Diffuse symmetric restricted diffusion involving the temporal, occipital, and parietal lobes as well as the insular cortex and minimally within the thalami. No significant edema or white matter involvement. These were thought to be related to PRES (posterior reversible encephalopathy syndrome) and not status epilepticus despite no corresponding involvement on T2 FLAIR. (**b**) Head CT after administration of ketamine and stopping of seizures on EEG. Diffuse loss of gray-white differentiation with effacement of cerebral sulci and basal cisterns consistent with diffuse cerebral edema.

## Data Availability

Not applicable.

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
