# Peer review of "Status Epilepticus in Post-Transplantation Hyperammonemia Involves Careful Metabolic Management"

_life, 2022, doi:10.3390/life12101471_

Round 1
Reviewer 1 Report
The authors present a case of post-lung transplant ALF with hyperammoniemia resulting in brain-death. There are some major issues with the case presentation:
- first of all, extensive English editing is needed in order to make the article clearer
- please include only the active substance not not the commercial drug name throughout the text
- it seems that that patients has many factors for ALF: drug toxicity, sepsis, haemodynamics (no data are available) and as such it fulfills the criteria for ALF definition according to AASLD (HE, bilirubin, coagulpathy??? - not reported)
- Why did the patient receive Ketamine in regard to brain edema?
Author Response
Thank you for your review of this article. As for your first comment, we have thoroughly gone through the entire text and edited it for grammar, spelling, and punctuation as well as sentence flow and structure to create a more polished manuscript. As for your second comment on drug names, we have included the generic names (active substance) for all the drugs mentioned. As for your third comment regarding diagnostic criteria for ALF, we have added the patient’s initial bilirubin and commented about its uptrend as well as the INR. The following was added to the text “Prior to HD40, INR had remained below 1.5 however between HD40 and HD60, INR was elevated to a average of ~3.0 before decreasing slightly to INR of ~2.0 from HD60 and onwards. Bilirubin has remained between 1 and 2 mg/dL until HD70 after which it began to rise precipitously to a peak of 18 mg/dL on HD 85”. Lastly, the patient received ketamine in an effort to end status epilepticus, not specifically for brain edema (the brain edema was largely a function of the hyperammonemic state).
Reviewer 2 Report
The authors described a case with hyperammonemia after lung transplantation. Regarding to management for post-organ transplantation, epilepticus is one of the major complications using immunosuppression. This hyperammonemia case report is a rare case and valuable one but this is not common situation that the authors concluded.
Author Response
Thank you for your time and review of this article. We appreciate the comment and have changed to text to reflect that this is not a common situation with the following text “SE has a wide variety of known causes however metabolic abnormality, especially hyperammonemia, is a rare etiologies”.
Reviewer 3 Report
1) The initial immunosuppression should be mentioned in detail. Figure 1 is not required for understanding. It would be better to depict a figure with tacrolimus dosage and trough levels as well as the addition of voriconazole that causes drug-drug-interaction. Tacrolimus trough level between 19 and 25 ng/ml is a differential diagnosis for neurological symptoms. There are plenty references regarding CNI neurotoxicty that should be cited somehow in this context and be discussed as well. 2) Did the patient had problems with sodium concentration (hyponatremia ?). 3) How was the the liver synthesis (INR, albumin), 4) You should mention, if the ammonia probes have been cooled always prior to analysis, otherwise the analysis is false positive
Author Response
Thank you for your time and review of this article.
- As per your comment on immunosuppression, the elevations in tacrolimus trough levels were transient and returned to levels below 8 ng/ml for the remainder of the hospitalization. We felt it was important to mention the patient’s immunosuppression in order to fully explain our differential diagnosis for this acute change in mental status however given the rapid reduction in tacrolimus levels upon implementing a significant dose reduction, we felt that this was not a contributing factor to his mental status change. We also thought CNI-related neurotoxicity to be less likely in this case as this is usually a reversible phenomenon that corrects rapidly with tacrolimus down-titration. We felt, however, that this could more so be contributing to the patient’s mild AKI. With concern for drug-drug interactions with respect to voriconazole, drug levels were monitored closely and tacrolimus was down-titrated immediately when seen to rise early in the admission. As recommended, key literature surrounding tacrolimus neurotoxicity has been discussed and cited in the text.
- The patient did not have any problems with serum sodium levels during this hospitalization. We have re-reviewed the patient chart and confirmed no evidence of clinically relevant hyponatremia or hypernatremia.
- We agree that further explanation of liver synthesis would be important in this case. The following was added to the text “Prior to HD40, INR had remained below 1.5 however between HD40 and HD60, INR was elevated to an average of ~3.0 before decreasing slightly to INR of ~2.0 from HD60 and onwards. Bilirubin has remained between 1 and 2 mg/dL until HD70 after which it began to rise precipitously to a peak of 18 mg/dL on HD 85”.
- The ammonia probes were always cooled, and the ammonia was elevated across multiple measurements as evidenced in Figure 2 making repeated false positives extremely unlikely. We have added a short few words about this in the text.